# Contribution of Axon Initial Segment Structure and Channels to Brain Pathology

**DOI:** 10.3390/cells12081210

**Published:** 2023-04-21

**Authors:** Juan José Garrido

**Affiliations:** 1Instituto Cajal, CSIC, 28002 Madrid, Spain; jjgarrido@cajal.csic.es; 2Alzheimer’s Disease and Other Degenerative Dementias, Centro de Investigación Biomédica en Red de Enfermedades Neurodegenerativas (CIBERNED), 28002 Madrid, Spain

**Keywords:** axon initial segment, brain disease, voltage-gated ion channels

## Abstract

Brain channelopathies are a group of neurological disorders that result from genetic mutations affecting ion channels in the brain. Ion channels are specialized proteins that play a crucial role in the electrical activity of nerve cells by controlling the flow of ions such as sodium, potassium, and calcium. When these channels are not functioning properly, they can cause a wide range of neurological symptoms such as seizures, movement disorders, and cognitive impairment. In this context, the axon initial segment (AIS) is the site of action potential initiation in most neurons. This region is characterized by a high density of voltage-gated sodium channels (VGSCs), which are responsible for the rapid depolarization that occurs when the neuron is stimulated. The AIS is also enriched in other ion channels, such as potassium channels, that play a role in shaping the action potential waveform and determining the firing frequency of the neuron. In addition to ion channels, the AIS contains a complex cytoskeletal structure that helps to anchor the channels in place and regulate their function. Therefore, alterations in this complex structure of ion channels, scaffold proteins, and specialized cytoskeleton may also cause brain channelopathies not necessarily associated with ion channel mutations. This review will focus on how the AISs structure, plasticity, and composition alterations may generate changes in action potentials and neuronal dysfunction leading to brain diseases. AIS function alterations may be the consequence of voltage-gated ion channel mutations, but also may be due to ligand-activated channels and receptors and AIS structural and membrane proteins that support the function of voltage-gated ion channels.

## 1. Introduction

Channelopathies refer to a group of disorders that arise from the dysfunction of ion channels. These channels are specialized proteins that regulate the movement of ions across the cell membrane, thereby controlling critical cellular functions such as electrical signaling and muscle contraction. Channelopathies can occur due to genetic mutations that affect the expression, function, and regulation of ion channels or their regulatory proteins, leading to a wide range of clinical manifestations [1]. However, reducing channelopathies to genetic mutations in ion channels underestimates the complexity of ion-channel-related pathologies, as the channel biology also depends on the membrane, cell location, and intercellular and extracellular biology. Further research is needed to better understand the genetic, molecular, and cellular mechanisms underlying channelopathies and to develop more effective therapies for these disorders.

Regarding brain channelopathies, a proper and coordinated action of voltage-gated ion channels and their distribution and density in neuronal domains determines the accurate function of brain networks [2]. Mutations in these channels are associated with brain diseases or mental disorders [3]; however, alterations of their density, location, or scaffold proteins are also a potential cause of brain diseases. Voltage-gated ion channels are highly concentrated at the axon initial segment (AIS) and are responsible for action potential generation. This concentration of voltage-gated ion channels is supported by a complex scaffold of membrane and cytoskeleton-associated proteins [4], whose regulation affects the channel density [5,6,7]. Moreover, the function of other ion channels or receptors modulates the AIS scaffold and voltage-gated ion channels [8,9], and is a source of AIS imbalance when their function or expression is altered.

### 1.1. Axon Initial Segment Structure and Physiology

Brain network functions depend on the coordinated action of multiple neurons and their regulation and support by glial cells. Each neuron in the network receives information through their dendritic tree and dendritic spines and processes and integrates this information in the soma. The changes in membrane potential arrive at an AIS, which then generate the corresponding action potentials to propagate the signal to the axon. Nodes of Ranvier are in charge of maintaining the signal until it arrives at the presynaptic terminal, which liberates neurotransmitters to produce a response in the next neuron. In this context, the AIS is the most reliable neuronal compartment, where action potentials are elicited [10,11]. An AIS acts as a gatekeeper controlling the output signal, and is characterized by a surprising structural plasticity, being able to modify its length, position, and composition to modulate the output signal when confronted with a deregulated input signal [12,13,14]. Furthermore, an AIS can also monitor the entry of proteins into the axon through the membrane diffusion barrier and control the cytoplasmic traffic, maintaining neuronal polarization [15]. In view of these important functions, it is easy to understand how the loss of AIS integrity or changes in their structure or structural plasticity leads to brain diseases or mental disorders. Moreover, the importance of AIS integrity for neuronal physiology is highlighted by the fact that AIS disruption precedes neuronal death [16].

### 1.2. Axon Initial Segment Composition and Structure

An AIS is a highly stable structure that is the first 20–60 μm of the axon (Figure 1). Its function as a membrane diffusion barrier is due to a high concentration of membrane proteins such as neurofascin, NrCAM, TAG1, ADAM22, and voltage-gated ion channels [15]. These membrane proteins are concentrated and tethered at the AIS through specific aminoacidic motifs that bind to AIS scaffold proteins such as ankyrinG, βIV-spectrin, or PSD-93 [17,18]. AIS proteins are organized into a complex network, which forms a diffusion barrier between the axonal and somatodendritic compartments of the neuron, thereby preventing the diffusion of proteins and lipids between these compartments [19]. Among them, ankyrinG and βIV-spectrin are essential in maintaining AIS structure and integrity, linking membrane proteins to the AIS cytoskeleton. AnkyrinG interacts with microtubules through EB proteins [20], while βIV-spectrin links ankyrinG to the actin cytoskeleton [21]. Both actin and microtubules at the AIS contain differential characteristics compared to the somatodendritic or axonal cytoskeleton. Microtubules at the AIS contain more detyrosinated and acetylated tubulin [22] and are more stable and act as a platform for axonal motor proteins [23] such as kinesin-1. Post-translational modifications (PTMs) of tubulin, such as acetylation, alter the ankyrinG distribution and impair kinesin-1 entry to the axon [6]. Tubulin PTMs confer a high degree of stability to the AIS, but additionally TRIM-46 (Tripartite motif containing 46) contributes to microtubule fasciculation [24]. Different actin structures have been identified at the AIS, forming actin rings and actin patches. Actin rings are supported by αII- and βIV-spectrin tetramers and are thought to participate in the initial axon structural plasticity [25], while actin patches may serve as a mechanism controlling the axonal entry of somatodendritic cargoes and their retrograde transport by dynein [26,27]. The AIS also contains an actin-related organelle, the cisternal organelle (CO), which is subject to structural changes associated with AIS plasticity during development and environmental changes [28,29]. Its function is unknown, but it was suggested to be putative Ca^2+^ store involved in calcium regulation at the AIS.

Among all AIS proteins, ankyrinG plays a crucial role at the structural and functional levels. AnkyrinG loss leads to AIS disassembly and consequently the loss of neuronal polarity and the appearance of dendritic spines at the former axon [30,31]. Functionally, the loss of ankyrinG impairs the concentration of sodium and some potassium voltage-gated ion channels, which contain an aminoacids sequence (AIS motif) for ankyrinG binding [32,33,34].

The complexity and specificity of the AIS structure, together with the still limited information available about its protein composition, renders it difficult to completely understand how action potentials are regulated and how voltage-gated ion channel density and location contribute to this regulation.

### 1.3. Voltage-Gated Ion Channels in the AIS

Voltage-gated ion channels are distributed in different axonal subdomains to carry out action potential initiation, conduction, and synaptic transmission. In addition, each neuronal type may have a different combination of channels in each neuronal domain [4,35]. Voltage-gated sodium channels (Na_v_1.2 and Na_v_1.6), which contribute to the increasing phase of action potential, are highly concentrated in the AIS and nodes of Ranvier [36], while presynaptic terminals contain mainly Na_v_1.2 channels. Na_v_1.1 channels are also tethered at the AIS of parvalbumin interneurons and Purkinje cells [37]. Their distribution in the AIS of pyramidal neurons is mainly proximal for Na_v_1.2, while Na_v_1.6 channels are distributed distally in the AIS [38]. Na_v_ channels are anchored at the AIS through the AIS motif in the intracellular II–III domain loop that interacts with the repeats domain of ankyrinG [32].

Different voltage-gated potassium channels (K_v_1.1, K_v_1.2, K_v_2.2, K_v_7.2, and K_v_7.3) are tethered at the AIS and contribute to the action potential spike duration and the control of resting membrane potential and threshold. K_v_7.2 and K_v_7.3 channels are also anchored to ankyrinG through a domain homolog to the AIS motif of sodium channels [33], while K_v_1.1 channels are anchored by the PSD-93 protein and are tethered distally at the AIS [18,39].

Less is known about the distribution and presence of voltage-gated calcium channels at the AIS. However, Ca_v_2.1 (P/Q-type) and Ca_v_2.2 N-type channels have been described at the AIS of L5 cortical pyramidal neurons [40] and Ca_v_3 (T- and R-type) channels in Purkinje neurons [41]. These channels regulate spike-timing, burst-timing, and the action potential threshold.

### 1.4. Hyperpolarization-Activated Cyclic Nucleotide-Gated Channels

HCN (hyperpolarization-activated cyclic nucleotide-gated) channels are ion channels that are expressed in various tissues throughout the body, including the heart, brain, and retina. These channels play an important role in regulating the excitability of cells by opening in response to hyperpolarization, allowing positive ions such as sodium and potassium to flow into the cell and depolarize it [42]. Among them, HCN1 is the only HCN channel identified at the AIS and is only present in auditory neurons [8], while HCN1, HCN2, and HCN3 are ubiquitously expressed in excitatory neurons [43] and their functional alterations or expression changes may contribute to brain pathology [44].

### 1.5. Neurotransmitter Receptors at the AIS

Voltage-gated ion channels can also be regulated by neurotransmitter receptors at the AIS. Serotonin 5-HT1A receptor is expressed at the AIS of auditory neurons and prefrontal pyramidal neurons [8,45] and dopamine D3R receptors modulate Ca_v_3.2 channels specifically at the AIS, suggesting their expression in the AIS [46]. Serotonin slows the activation/deactivation kinetics of ion channels, modulating HCN channels in auditory neurons via 5-HT1A receptors which are coupled to the G proteins (Gi/Go) that decrease the cyclic adenosine monophosphate (cAMP) concentration [47]. The activation of 5-HT1A receptors also reduces motoneuron excitability [48]. However, the exact role of serotonin regulation of the AIS is not completely understood. Dopamine D3R receptors are also coupled to Gi/Go proteins and modulate the calcium influx through AIS-localized Ca_v_3 calcium channels [9]. Their function is to control the overall neuronal activity by regulating the excitability of the spike initiation zone.

### 1.6. Axon Initial Segment Plasticity and Modulation

The AIS has a surprising capacity to adapt its composition, length, and location in response to modifications of the inputs received from other neurons (Figure 2). Thus, the action potential generation and its threshold depend not only on voltage-gated ion channels at the AIS but also on changes in other channels or receptors of the AIS. AIS plasticity was first described in chicken auditory neurons and cultured mouse hippocampal neurons [13,49]. Afferent deprivation of nucleus magnocellularis neurons (NM) induces AIS elongation [49], while increased extracellular potassium levels induces a movement of AIS a few microns away from the soma in cultured hippocampal neurons [13]. AIS elongation due to auditory deprivation increases the Na^+^ current in the axon and enhances the excitability of the neurons, thereby contributing to a compensation for the loss of auditory nerve activity. Besides AIS elongation, auditory deprivation also contributes to decreased K_v_1.1 expression and increased K_v_7.2 expression and redistribution along the AIS of both channels [50]. These changes are completed around 7 days after deprivation. In contrast, chronic depolarization increases extracellular potassium, as occurs in brain injury, strokes, and other brain disorders, and moves AIS away from the soma of cultured hippocampal excitatory neurons after 2 days [13]. This plasticity does not occur in GABAergic hippocampal neurons and is inverted, proximally relocated, and lengthened in OB dopaminergic interneurons [51]. AIS plasticity prevented the blocking of voltage-gated calcium channels (VGCCs) with nifedipine, implying that calcium increases due to L-type VGCCs may therefore contribute to AIS relocation [13]. Different calcium-dependent mechanisms may control AIS plasticity. While AIS relocation in excitatory neurons depends on calcium-dependent phosphatase calcineurin, no effect on AIS plasticity was detected in OB dopaminergic neurons after calcineurin inhibition [51]. The calcium-calpain-mediated mechanism also participates in AIS plasticity and integrity. Calpain activation proteolyzes voltage-gated sodium channel VGSCs [52], ankyrinG, and αII-spectrin [53] and its inhibition protects AIS integrity [12,16]. One of the mechanisms that promotes calpain activation is mediated by the purinergic receptor P2X7, which allows calcium entry through its pores and activates calpain, decreasing ankyrinG and voltage-gated sodium channel densities in the AIS in mouse cortical and hippocampal neurons [12]. Good calcium regulation inside the AIS remains elusive. The cisternal organelle (CO) may play a role in this calcium regulation, as it is enriched with ryanodine receptors (RyRs) [54] for the release of calcium ions, which then activate calcium-dependent processes within the AIS [55]. CO is thought to be a specialized form of the smooth endoplasmic reticulum (ER) and plays a crucial role in regulating the excitability of neurons and AIS plasticity.

In addition to its role in calcium signaling, the cisternal organelle has also been implicated in the regulation of other ion channels within the AIS. For example, one study found that the cisternal organelle is responsible for the sequestration of voltage-gated potassium channels (K_v_2.1) within the AIS [54]. This sequestration allows for the proper localization of K_v_1.2 channels within the AIS, which is crucial for the regulation of action potential firing. The relationship between the cisternal organelle and AIS modulation is still not fully understood and further research into the cisternal organelle may lead to a better understanding of AIS structural plasticity and the pathophysiology of several neurological and psychiatric disorders, including epilepsy, autism spectrum disorder, and schizophrenia.

The changes in the structural and functional properties of the AIS enable neurons to efficiently manage various physiological or pathological environments and maintain brain network homeostasis. Recent works has highlighted a central role of the AIS in the homeostatic regulation of neuronal input–output relations in physiological conditions. Long-term sensory deprivation elicits an increase in the AIS length, accompanied by an increase in neuronal excitability, while sensory enrichment results in a rapid AIS shortening [56]. Although it is easy to understand how changes in protein density may occur in the AIS, the mechanisms involved in length and location changes are mostly unclear. Recent studies suggest that structural and functional refinements are coordinated and regulated differentially but work synergistically to optimize the neuronal output [57]. In addition to calcium-dependent mechanisms, other structural and functional proteins regulate the AIS by modulating microtubules and the actin cytoskeleton, or even other neuronal compartments that influence AIS characteristics. Alterations in actin dynamics through pharmacological depolymerization or stabilization of actin microfilaments are known to influence action potential properties and the sodium channel density, increasing the sodium current after actin depolymerization [11]. However, the mechanisms regulating actin dynamics in the AIS are not well known. mDia1 formin is an actin filament regulator, whose inhibition or partial depletion in neurons leads to a lower density of sodium channels in the AIS and changes in the action potential parameters (rheobase and threshold) [7]. mDia1 may participate in AIS modulation through actin mechanisms regulating AIS length and microtubules mechanisms regulating ankyrinG density. A reduced sodium channel density was also observed in cultured neurons after P2Y1 receptor inhibition or depletion, which decreased actin patch density and p-myosin light chain (pMLC) expression at the AIS [58]. This complex is associated with actin rings in the AIS, providing a mechanism for activity-dependent structural plasticity [59]. AIS microtubules contain posttranslational modifications, such as acetylation or detyrosination, that confer the specific characteristics to the AIS. The loss of this specificity leads to AIS disruption and the reduction in voltage-gated channel concentration, as happens after tubulin deacetylase HDAC6 inhibition [6]. A better knowledge of the coordination of AIS cytoskeleton regulatory elements may lead to a better understanding of how to modulate AIS structural plasticity. A recent work demonstrated that exposure of neurons to a high K^+^ medium leads to AIS shortening due to Ca^2+^-related mechanisms that lead to cdk5 kinase and microtubule disassembly [60]. In this context, a closely related kinase, GSK-3, has been found at the AIS and its inhibition leads to lower ankyrinG density at the AIS [5].

The structural and functional complexity of the AIS, together with its plasticity to maintain neuronal function into physiological parameters, highlights its role in circumventing potential pathological alterations.

## 2. Intrinsic and Extrinsic Axon Initial Segment Factors Contributing to Channelopathies

The function of voltage-gated ion channels in the AIS depends on the supporting scaffold and regulatory and interacting proteins at the AIS (intrinsic factors), but also is coordinated with environmental and physiological alterations in the somatodendritic domain, presynaptic terminals, and supporting glial cell functions and interactions (extrinsic factors).

### 2.1. AIS Intrinsic Factors

#### 2.1.1. AIS Voltage-Gated Ion Channels Mutations

Different brain channelopathies due to mutations in sodium, potassium, or calcium voltage-gated channels expressed at the AIS have been described. These mutations alter neuronal excitability, modifying channels’ activation/inactivation parameters and generating mainly epilepsies, pain, or myotonias [61]. Mutations in several voltage-gated sodium channel genes, including Na_v_1.1, Na_v_1.2, or the sodium channel beta-1 subunit, have been associated with GEFS+ (generalized epilepsy with febrile seizures plus). These mutations can result in gain-of-function or loss-of-function of Na_v_ channels, leading to altered excitability and synaptic transmission in the brain [62]. Among them, Dravet syndrome is a severe form of epilepsy that usually begins in infancy and is characterized by prolonged seizures, developmental delays, and cognitive impairment. The majority of Dravet syndrome cases are caused by mutations in the *SCN1A* gene, which encodes the Na_v_1.1 channel. These mutations usually result in loss-of-function of Nav1.1 channels [63], leading to reduced excitability and impaired action potential generation in inhibitory interneurons, which in turn can disrupt the balance of excitatory and inhibitory signaling in the brain and increase seizure susceptibility.

Mutations in the *KCNQ2* and *KCNQ3* genes, which encode K_v_7.2 and K_v_7.3 channels, respectively, have been identified as a cause of BFNE (benign familial neonatal epilepsy). These mutations can lead to loss-of-function of K_v_7 channels, resulting in reduced repolarization of the membrane potential and increased excitability of neurons in the brain [64]. Besides, K_v_7.2/7.3-related seizures may be the result of an activity-dependent increase in *KCNQ* gene transcription in hyperexcitable neurons [65], leading to changes in K_v_7 channel expression during development and the disappearance of spontaneous seizures in adult BFNS patients. Developmental changes in K_v_7 channel subcellular location may also contribute to epilepsies. K_v_7 channels appear progressively during postnatal development and change their location from soma to axonal domains [66], which may explain the disappearance of neonatal epilepsies in adults.

More than 40 different mutations have been identified in the K_v_1.1 channel, as it is the only gene related to episodic ataxia type 1 [67]. K_v_1.1 loss-of-function leads to increased spontaneous spike-wave discharges. Neuronal activity can also modulate the K_v_1.1 density at the AIS, shifting from AIS to soma after activity deprivation [68]. Additionally, as it will be mentioned later, K_v_1.1 is involved in other neurological disorders mediated by other interacting proteins.

Voltage-gated calcium channel mutations have been detected in a variety of neurological disorders [69]. Mutations in the Ca_v_2.1 channel are associated with ataxias [70] and some forms of epilepsy [71]; however, no mutations have been found in other Ca_v_2 channels. All three Ca_v_3 channel mutations have been identified in autism spectrum disorder (ASD) [72] and amyotrophy lateral sclerosis (ALS) [73], but their association needs to be validated. Rare Ca_v_3.3 mutations have been confirmed as a factor that reduce the disease risk burden in schizophrenia [74].

In this context, it is important to state that although all these channels are located in the AIS and are thought to function in the AIS, further research is necessary to ascertain that their dysfunction in the AIS is responsible for observed pathologies. While these channels’ concentrations in the AIS is higher, it is not possible to exclude that their expression in other neuronal compartments can have an important role in described pathologies.

#### 2.1.2. Voltage-Gated Ion Channels Scaffold Proteins at the AIS

Voltage-gated ion channels need the support of scaffold proteins to maintain their density and membrane expression in the AIS. In fact, mutations in ankyrinG and spectrins are related to brain channelopathies affecting ion channel density, distribution, or function.

Mutations in ankyrinG contribute to diseases by disrupting ion channel clustering in the AIS and nodes of Ranvier [75]. Mutations and SNPs (single nucleotide polymorphims) in the *ANK3* gene that code for ankyrinG are related to bipolar disorder [76], schizophrenia [77], post-traumatic stress disorder (PTSD) [78], attention-deficit hyperactivity disorder (ADHD), autism spectrum disorders, or intellectual disability (ID) [79,80]. However, the mechanisms by which risk SNPs affect the expression/function of ankyrinG and induces disease are not well understood.

αII spectrin and βIV spectrin mutations are also related to early infantile epileptic encephalopathy-5, congenital hypotonia, developmental delay, and intellectual disability from early childhood [81]. αII spectrin mutations leads to reduced Nav channel clustering and the absence of action potential firing [82]. Moreover, the dendritic and axonal complexity are reduced in neurons expressing human αII spectrin variants [83]. Most βIV spectrin mutations affect their binding capacity to ankyrinG, compromising AISs structural stability [84,85].

### 2.2. AIS Extrinsic Factors

Besides mutations and SNPs in AIS scaffold proteins, other mechanisms acting in other neuronal domains can affect their density in the AIS, which in turn affect the density of sodium and potassium channels in the AIS, generating alterations in neuronal excitability.

As mentioned above, P2X7 receptor activation opens the receptor and allows the entry of calcium in neurons activating calpain, which in turn proteolyzes ankyrinG, βIV-spectrin, and Nav channels [12]. Interestingly, P2X7 is not expressed in the AIS and is only detected in axon terminals and dendrites [86]. The expression of P2X7 receptors increases due to brain injury and neuroinflammatory brain diseases [87]. P2X7 inhibition increases the Na^+^ current amplitude and its activation reduces it and decreases neuronal excitability in cortical and hippocampal neurons [12]. However, the P2Y1 receptor is necessary for AIS initial development and ankyrinG tethering in the AIS [58]. Both purinergic receptors function in an antagonistic way, controlling cAMP levels during initial axon development [86]. Therefore, a balance between these two purinergic receptors may serve as a sensor of purines that modulate AIS protein density. A second demonstration of the role of calcium dynamics and the role of receptors outside the AIS that influence AIS composition and function is Na/K-ATPase, which shows increased expression in CA1 hippocampal neurons in Angelman syndrome (AS) mice [88]. Angelman syndrome occurs due to the loss of function of the maternal ubiquitin-protein ligase E3A (UBE3A) gene located in the 15q11.2–13.3 region and is characterized by the appearance of epilepsy, motor abnormalities, intellectual disability, or autism [89]. A mutant mouse carrying this deletion (AS mice) is characterized by increased ankyrinG, Na_v_1.6 channel, and α1 subunit of Na/K-ATPase (α1-NaKA) densities in CA1 hippocampal neurons [88]. This increased expression was associated with a significantly lower dendritic Ca^2+^ influx in hippocampal CA1 pyramidal neurons of AS mice, and Na/K-ATPase inhibition was sufficient to restore Ca^2+^ dynamics [90].

The roles of these channels outside the AIS highlights the influence that other neuronal domains have on the AIS and the density and location of AIS voltage-gated ion channels. In fact, a mathematical model was proposed illustrating how changes in dendritic arbor complexity change the AIS structural plasticity [91]. In hippocampal neurons, inhibition or suppression of the cannabinoid receptor, CB1, decreases dendritic growth and concomitantly decreases ankyrinG density during early AIS development [92]. Moreover, the apical dendrite diameter is also correlated with the AISs distance to the soma [93]. Activation of dendritic receptors, such as NMDARs, contributes to K_v_7.2/7.3 channel endocytosis and degradation by calpain [94].

## 3. Axon Initial Segment, Mental Disorders, and Neurodegenerative Diseases

Several pieces of evidence suggest that alterations in the structure and function of AIS contribute to the pathophysiology of various psychiatric, neurodevelopmental, neurodegenerative, or autoimmune diseases, as well as brain trauma or injury. While this review will focus on a subset of brain diseases, other diseases (i.e., type 2 diabetes or neuropathic pain) are also related to changes in AIS composition and plasticity [95,96,97].

### 3.1. Mental Disorders

Bipolar disorder is a mental health condition characterized by episodes of manic and depressive symptoms. The exact causes of bipolar disorder are not fully understood, but it is believed to involve a complex interplay of genetic, environmental, and neurobiological factors. Research has suggested that abnormalities in the function and structure of neurons, including those in the AIS, may contribute to the development of bipolar disorder. Gene array studies in bipolar disease patients have shown decreased expression of potassium channel K_v_1.1, while K_v_7.2 and K_v_7.3 channels had increased expression [98]. Different studies have highlighted the role of *ANK3* gene single nucleotide polymorphims (SNPs) as a risk factor for bipolar disorder. These SNPs affect ankyrinG expression and concomitantly the proper anchoring of potassium channels [99]. Moreover, *ANK3* mRNA can be regulated by the expression level of microRNAs in bipolar disorder patients, such as miR34a and miR10b-5p [100]. ankyrinG (480 kDa) is essential for AIS assembly and three missense mutations have been identified in humans. These mutations increase the AIS length, decrease the ankyrinG density, and prevent βIV spectrin recruitment. As a consequence, voltage-gated sodium channels are diffused along the longer AIS and action potential firing has a reduced temporal precision [101]. Furthermore, fore-brain specific knockout of ankyrinG in adult mice shows characteristics reminiscent of aspects of human mania [102].

Schizophrenia is a group of complex mental disorders characterized by psychotic, negative, and cognitive symptoms. Among other alterations, some studies have shown that individuals with schizophrenia have alterations in the expression and distribution of sodium channels in various regions of the brain, including the prefrontal cortex and hippocampus [103]. Loss of function (LoF) mutations and rare coding variants (RCVs) in sodium channels increase the risk of schizophrenia [103]. Mouse models related to schizophrenia have shown increased Na_v_1.2 and Na_v_1.6 channel expressions and changes in the electrophysiological properties of action potentials [104,105]. There is no clear evidence of the role of other potassium or calcium channels expressed in the AIS in schizophrenia. The link between *ANK3* and schizophrenia remains unclear. Some GWAS studies have proposed an association between *ANK3* SNPs and schizophrenia; however, a large GWAS did not show a significant association [106].

### 3.2. Neurodevelopmental Disorders

Neuronal polarization, functional polarity, and brain network physiology depend on proper early AIS development and maturation. Several neurodevelopmental disorders have been related to early alterations in AIS development. The mouse model of Angelman syndrome exhibited a significant increase in the AIS length in hippocampal neurons [88]. This syndrome is due to the loss of function of the *UBE3A* gene and is associated with intellectual disability, epilepsy, ataxia, and autism. An increased length is accompanied by a higher expression of voltage-gated sodium channels and ankyrinG. The same length increase has been detected in a fragile X mouse model (Fmr1^−/y^) that shows increased neuronal excitability [107]. More recently, the loss of a transcription factor, Pax6, was found to change the length and position of the AIS in prethalamic neurons [108]. Similarly, the loss of function of Rbfox proteins that regulate alternative splicing and posttranscriptional regulation leads to defects in AnkG localization and an immature electrophysiology [109], showing the importance of AnkG developmental splicing. In fact, neurons that express ankyrinG, including a 33 nucleotide cassette upstream of the first ZU5 domain, do generate significantly less action potentials, indicating a lower density of voltage-gated ion channels [101].

Autism spectrum disorders are also related to AIS alterations. Subjects with autism show a decreased GABA synthesis in the prefrontal cortex that can lead to downregulation of the GABAARa2 protein in the AIS of pyramidal neurons and affect the inhibitory control of action potentials, contributing to an excitation/inhibition imbalance [110]. Autism mouse models also show a shortening of the AIS in the primary somatosensory barrel cortex, similar to the AIS length change in the same region in ADHD (attention-deficit hyperactivity disorder) model rats [111]. The contribution of voltage-gated ion channels or other scaffold proteins to autism has been proposed, but results only allow to suggest that they contribute to an increased susceptibility to autism spectrum disorders.

### 3.3. Neuroinflammatory Diseases and Glial Cells

Neuroinflammatory diseases are a group of conditions that involve inflammation in the central nervous system (CNS). Inflammation is the normal response of the immune system to injury or infection, but in some cases, it can become chronic and contribute to the development of various neurological disorders. Neuroinflammatory diseases include, among others, multiple sclerosis (MS), amyotrophic lateral sclerosis (ALS), and also those derived from a brain trauma or brain stroke. Moreover, glial cell activation or alterations contribute significantly to this group of brain diseases.

Several studies have highlighted structural and functional changes in the AIS after brain damage due to blast wave, oxygen/glucose deprivation, or brain ischemia. Exposure to a blast wave generates a traumatic brain injury, leading to altered cognition, memory, and behavior. Translated to a rat model, the consequences are a shortening of the AIS and decreased neuronal excitability [112]. While the cellular and molecular changes are unknown, other models of brain injury have allowed us to understand several potential mechanisms. An oxygen and glucose deprivation model representative of brain ischemia allowed us to determine that sodium channels, structural proteins, and membrane proteins are proteolyzed by calpain [16]. As mentioned above, one of the mechanisms that activates calpain in the AIS is mediated by the purinergic P2X7 receptor. This receptor is not located in the AIS but its inhibition protects AIS voltage-gated ion channels and AIS structural proteins protecting the AIS integrity after middle cerebral artery occlusion (MCAO)-induced ischemia [12].

ALS and MS are characterized by alterations in nerve impulse conduction. In addition to myelin sheath alterations, models of both diseases show alterations in voltage-gated ion channel expression in the AIS and AIS plasticity. Studies have shown that the expression of Na_v_1.6 channels is increased in demyelinated axons in MS patients, activating the accumulation of intra-axonal calcium through the Na^+^/Ca^2+^ exchanger, which may contribute to the development of axonal dysfunction and neurodegeneration [113]. Motoneuron AISs in an ALS mice model G127X SOD1 were longer and thinner during the symptomatic stage and results suggest that the Na_v_1.6 channel expression increases [114]. However, G127X SOD1 mice show shorter AISs before the symptomatic stage, suggesting that AIS plasticity mechanisms may be a target for ALS. Other ALS models show reduced selective transport of cargoes before pathology appeared, suggesting AIS dysfunction [115].

A recent study on multiple sclerosis human tissue has shown that the gap between soma and AIS increases in pyramidal neurons and Purkinje cells in multiple sclerosis tissue and that the AIS is longer in cortical neurons of cortical inactive lesions [116]. Whether these AIS plasticity changes lead to voltage-gated ion channel expression or function alterations remains elusive, and further studies are necessary to decipher the potential role of the AIS in ALS and MS.

Glial cells play an important role in brain physiology and neuronal modulation. Astrocytes and microglia play a role in ischemia, ALS, or MS; however, their potential relation with the AIS needs further investigation. A small percentage of AISs in the cortex are associated and contact with microglial cells, and interestingly activated microglia lost its interaction with the AIS [117]. This interaction seems to have a physiological role and needs an intact AIS to be preserved. The role of this microglia–AIS contact in the regulation of voltage-gated ion channels is unknown; however, it is essential for chandelier cell GABAergic synaptogenesis in the AIS of neocortical pyramidal neurons [118].

Numerous studies point towards a crucial role of astrocytic ion channels linking the CNS microenvironment to astroglial responses, modulating health, inflammation, and ischemia. Even if astrocytes are non-excitable cells, these cells express a battery of potassium and calcium voltage-gated ion channels, as well as other ion channels that act as sensors and modulators [119]. Increased intracellular Ca^2+^ promotes ATP release from astrocytes, which modulates neuronal excitability and decreases the conduction speed in myelinated axons through their action on A2aR receptors and HCN2 channels in the AIS and nodes of Ranvier [120].

The role of microglia and astrocytes in brain diseases related to neuronal excitability and survival is increasingly studied; however, how these cells can participate in disease onset regulating their own ion channels and those in neurons remains unknown.

### 3.4. Autoimmune Diseases

A group of autoimmune diseases have been characterized as affecting structural and functional proteins in the AIS. Autoantibodies to βIV spectrin and TRIM46 have been detected in paraneoplastic CNS disorders [121,122,123]. AnkyrinG autoantibodies have been identified in some people affected by viral infections or suffering from cancer. AnkyrinG autoantibodies have been identified in AIS, nodes of Ranvier, and the subpial space in a patient living with human immunodeficiency virus (HIV) suffering steroid-responsive meningoencephalitis [124] and in a patient with a diagnosis of metastatic ovarian cancer and seizures [124]. Autoantibodies to Lgi1 are characteristic of limbic encephalitis [125]. Lgi1 interacts with the K_v_1.1 channel and is required to modulate the K_v_1.1 density in the AIS. Lgi1 knockout mice show increased neuronal excitability and suffered from seizures [126]. Together with Lgi1, Caspr2 also interacts with the K_v_1.1 channel, and the appearance of Caspr2 autoantibodies is associated with autoimmune encephalitis [127]. Autoantibodies to voltage potassium channels are also related to Morvan’s syndrome and have been found in schizophrenia patients [128].

### 3.5. Alzheimer’s Disease

Recent studies have proposed an excitatory/inhibitory imbalance as a potential onset mechanism for Alzheimer’s disease. Actually, voltage-gated sodium channels and ankyrinG expression was significantly decreased in an Alzheimer’s disease mouse model APP/PS1 [129] from postnatal stages. Young APP/PS1 mice showed a reduced number of action potentials, while older APP/PS1 mice have an increased number of spikes and hyperexcitability [130]. This maybe the consequence of an increased number of cell surface Na_v_1.6 channels at around 7 months [131]. In fact, Na_v_1.6 knockdown is capable of ameliorating cognitive defects and reducing hyperexcitability in APP/PS1 mice [132]. Meanwhile, there are no data regarding K_v_ channel expression at the AIS in Alzheimer’s mice models, even though interaction of K_v_7 channels with BACE1 was described [133]. Altered calcium homeostasis in neurons or glial cells is an early event observed in mouse models of Alzheimer disease [134]. Regarding the AIS, the potential role of the cisternal organelle in Ca^2+^ homeostasis deregulation has not been studied; however, the presence of ryanodine receptors coupled to Ca_v_3 channels in the AIS [55] suggests the role of these channels in action potential deregulation.

Moreover, AIS shortening was identified in Alzheimer’s disease mouse models or patients [135,136,137]. The absence of AIS structural plasticity was demonstrated in a frontotemporal dementia model due to a tau protein mutation identified in humans and its relation to EB1/EB3 proteins and microtubules in the AIS [138].

In conclusion, the dysregulation of voltage-gated ion channels can have profound effects on neuronal function and contribute to the development of mental disorders and neurodegenerative diseases. Understanding the role of these channels in disease pathogenesis may lead to the development of new treatments and therapies for these devastating conditions. However, it is important to take into account that voltage-gated ion channels pathologies are not only due to alterations in channel structure or expression, but also due to changes in the neuronal domain where their function is exerted and changes in their scaffold and interacting proteins.

## Figures and Tables

**Figure 1 cells-12-01210-f001:**
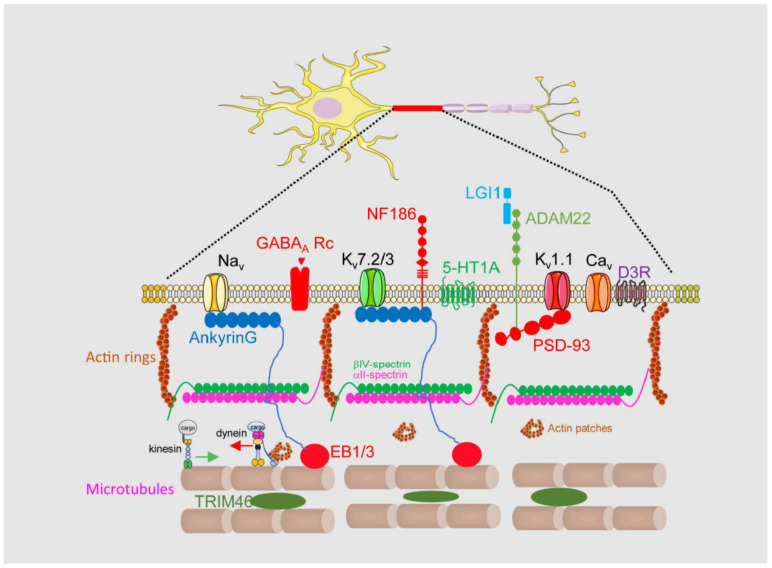
Axon initial segment structural and functional composition. The AIS is located in the first 20 to 60 μm of the axon and has a differentiated membrane composition with regard to somatodendritic and axonal domains. It is characterized by a high concentration of voltage-gated sodium (Na_v_) and potassium (K_v_) channels anchored to ankyrinG or PSD-93 scaffold proteins. AnkyrinG links ion channels to microtubules through EB1/3 proteins and to actin cytoskeleton through its binding to βIV-spectrin. The AIS contains also voltage-gated calcium channels (Ca_v_), serotonin, dopamine and GABA_A_ receptors. TRIM46 binds microtubules and contribute to their fasciculation. Their special characteristics allows the control of axonal cargoes entry in the axon.

**Figure 2 cells-12-01210-f002:**
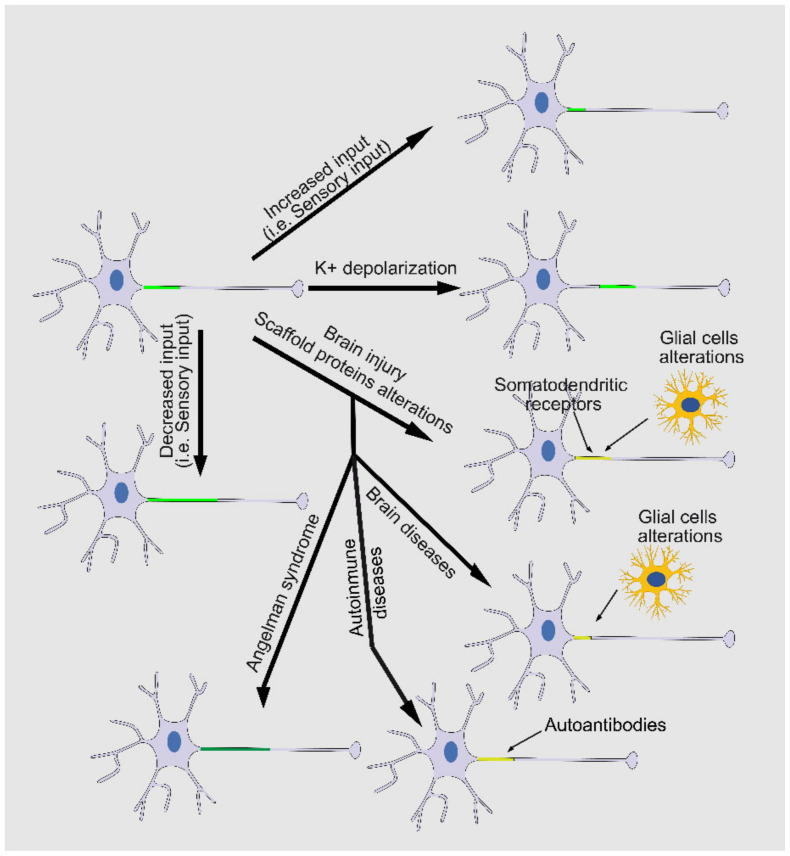
Axon initial segment plasticity in response to intrinsic and extrinsic factors. The AIS has significant plasticity and adapts to intrinsic and extrinsic alterations in order to maintain neuronal homeostasis. AIS plasticity could be due to alterations in the signals received from other neurons and sensory structures, intrinsic alterations in its structural and functional composition, or external signals coming from glial cells or autoantibodies against AIS proteins. AIS protein density is shown in green (high density), light green (normal density), or yellow-green (low density).

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
