# Peer review of "Contribution of Axon Initial Segment Structure and Channels to Brain Pathology"

_cells, 2023, doi:10.3390/cells12081210_

Round 1

Reviewer 1 Report

Garrido has written an interesting and comprehensive review on the contribution of AIS ion channels and scaffolds to disease.  The review is well written and includes all the major players and published observations. I think this is a nice contribution to the literature and will be of interest to a broad spectrum of neuroscientists.  I have only a few suggestions for improvement/correction:

1.     Lines 204-213:  this should be Kv2.1, not Kv1.2

2.     Section 2 – I think it is important for the author to explicitly state that although all these channels are located at the AIS and are thought to function at the AIS, we do not know if their dysfunction at the AIS is responsible for the observed pathologies.  These channels are also found in many other locations in the neuron – perhaps at lower densities in some instances, but we cannot exclude a greater role played by those channels than what is found at the AIS.

3.     Line 298: codifies should be codes

4.     Line 316: proteolysis should be proteolyzes

5.     Line 365:  “Furthermore fore-brain specific knockout of AnkG in adult conditional…”

6.     Line 391: inmature should be immature

7.     Section 3.4 on autoimmune diseases:  there are two new papers on autoantibodies against b4 spectrin and AnkG – both from the lab of Sam Pleasure that would be appropriate to include.  Especially the one reporting autoantibodies against AnkG.

8.     There are several other typographical or grammatical errors I encourage the editor to catch in the proof stage.

minor grammar corrections

Author Response

Reply to Reviewer 1

Garrido has written an interesting and comprehensive review on the contribution of AIS ion channels and scaffolds to disease.  The review is well written and includes all the major players and published observations. I think this is a nice contribution to the literature and will be of interest to a broad spectrum of neuroscientists.  I have only a few suggestions for improvement/correction:

I thank the reviewer for this constructive review. I have modified text according to suggestions and corrected typographical errors. Text modifications are highlighted in yellow in the revised manuscript and any change in the manuscript can be followed by “Track Changes” function in MS word. 

  1. Lines 204-213:  this should be Kv2.1, not Kv1.2 (Modified)
  2. Section 2 – I think it is important for the author to explicitly state that although all these channels are located at the AIS and are thought to function at the AIS, we do not know if their dysfunction at the AIS is responsible for the observed pathologies. These channels are also found in many other locations in the neuron – perhaps at lower densities in some instances, but we cannot exclude a greater role played by those channels than what is found at the AIS.

I have added a paragraph (lines 297-301) to state this important idea. It would be interesting to understand in which proportion voltage gated ion channels at the AIS are involved in disease.

  1. Line 298: codifies should be codes (Modified)
  2. Line 316: proteolysis should be proteolyzes (Modified)
  3. Line 365: “Furthermore fore-brain specific knockout of AnkG in adult conditional…”  (Modified)
  4. Line 391: inmature should be immature

I have checked for typographical errors using English (EEUU) dictionary in Word, and modified according.

  1. Section 3.4 on autoimmune diseases: there are two new papers on autoantibodies against b4 spectrin and AnkG – both from the lab of Sam Pleasure that would be appropriate to include.  Especially the one reporting autoantibodies against AnkG.

I have added this information. Thanks for putting the emphasis on it, as it is important to note that other diseases (infections or cancer) can also modify aspects of the AIS.

  1. There are several other typographical or grammatical errors I encourage the editor to catch in the proof stage.

I have checked for typographical errors and tried that grammar is correct. Hope I got it right.

Reviewer 2 Report

This manuscript thoroughly reviews the structure, plasticity, and function of the axon initial segment (AIS). In the first section, the author describes structure and molecular composition at the AIS such as voltage-gated sodium channels and potassium channels, ankyrinG, and αII/βIV spectrins (Figure 1), and the structural plasticity of AIS (length and location) in response to the changes in the inputs from other neurons (Figure 2) and its underlying molecular mechanisms. In the second section, the author reviews the AIS factors (e.g. ion channel mutations) that contribute to channelopathies in detail. Finally, in the third section, the author reviews the pathophysiology related to the AIS alterations leading to brain dysfunction in a wide variety of diseases and conditions such as mental disorders. Overall, this review article is very well written. Since emerging evidence indicates the importance of AIS in health and disease, this review will be helpful for many researchers in this field. Please consider minor points listed below to further improve this manuscript.

1. Structural plasticity of AIS (length and location) is one of the hottest fields in neuroscience as discussed in this manuscript (Figure 2, autism spectrum disorders (ref. 106), traumatic brain injury (ref. 107), multiple sclerosis (ref. 111), etc.). Therefore, it would be better to include recent observations in this aspect in health and disease (there are many articles in the literature): for example, homeostatic AIS plasticity (Jahan et al., J Neurosci 2023; Jamann et al., Nat Commun 2021; etc.). In section 3.5 Alzheimer’s disease, discussion of AIS structural changes would be helpful (e.g. Ma et al., J Neurosci 2023; Anton-Fernandez et al., Sci Rep 2022; etc.). Also, AIS structural changes are reported in the conditions related to Alzheimer’s disease or associated with cognitive impairment such as frontotemporal dementia, aging, type 2 diabetes, neuropathic pain, etc.

2. Page 2, line 46: “axon initial segment (AIS)”, then “AIS” should be used for the rest of the manuscript.

3. In Figure 2, some labels are too small.

Author Response

Reply to Reviewer 2

This manuscript thoroughly reviews the structure, plasticity, and function of the axon initial segment (AIS). In the first section, the author describes structure and molecular composition at the AIS such as voltage-gated sodium channels and potassium channels, ankyrinG, and αII/βIV spectrins (Figure 1), and the structural plasticity of AIS (length and location) in response to the changes in the inputs from other neurons (Figure 2) and its underlying molecular mechanisms. In the second section, the author reviews the AIS factors (e.g. ion channel mutations) that contribute to channelopathies in detail. Finally, in the third section, the author reviews the pathophysiology related to the AIS alterations leading to brain dysfunction in a wide variety of diseases and conditions such as mental disorders. Overall, this review article is very well written. Since emerging evidence indicates the importance of AIS in health and disease, this review will be helpful for many researchers in this field. Please consider minor points listed below to further improve this manuscript.

  1. Structural plasticity of AIS (length and location) is one of the hottest fields in neuroscience as discussed in this manuscript (Figure 2, autism spectrum disorders (ref. 106), traumatic brain injury (ref. 107), multiple sclerosis (ref. 111), etc.). Therefore, it would be better to include recent observations in this aspect in health and disease (there are many articles in the literature): for example, homeostatic AIS plasticity (Jahan et al., J Neurosci 2023; Jamann et al., Nat Commun 2021; etc.). In section 3.5 Alzheimer’s disease, discussion of AIS structural changes would be helpful (e.g. Ma et al., J Neurosci 2023; Anton-Fernandez et al., Sci Rep 2022; etc.). Also, AIS structural changes are reported in the conditions related to Alzheimer’s disease or associated with cognitive impairment such as frontotemporal dementia, aging, type 2 diabetes, neuropathic pain, etc.

Thanks very much for the constructive review. I have taken in account suggestions and added more information including the works mentioned by the reviewer. The text has been modified in section 2 (lines 214-217 and lines 241-245) and section 3.5 (lines 504-507). In addition, lines 357-361 now mention other diseases such as aging, type 2 diabetes or neuropathic pain.

  1. Page 2, line 46: “axon initial segment (AIS)”, then “AIS” should be used for the rest of the manuscript.

I have modified text and all, except the first mention to axon initial segment, are modified to AIS. I have considered that keeping the entire name in titles was important.

  1. In Figure 2, some labels are too small.

Figure 2 has been modified to increase labels and make it easier to follow.

A graphical abstract will be provided with the manuscript. Due to the dimensions, I have only incorporated general lines of all the possibilities that the AIS offer to neuronal plasticity in physiology and pathology.

Again, thanks very much for your review contribution to make a better review.
